# OpenReview forum: "HackWorld: Evaluating Computer-Use Agents on Exploiting Web Application Vulnerabilities"
_ICLR.cc/2026/Conference — ICLR 2026 Poster_

### Official Review · Reviewer_mq7d · 2025-10-31

**Soundness:** 3
**Presentation:** 3
**Contribution:** 3
**Rating:** 6
**Confidence:** 3

**Summary:**

This paper develops a test environment for evaluating the ability of agent AI systems to navigate websites and detect security vulnerabilities. The test environment is interesting in itself. It involves 36 curated applications spanning 11 frameworks and 7 languages, with realistic vulnerabilities including injection flaws, authentication bypasses, and unsafe input handling.The paper also evaluates a set of agents against this test suite and finds them to have limited power to uncover sophisticated vulnerabilities. Specifically, the exploitation rates are below 12%, often struggling to plan multi-step attacks and use security tools effectively. The paper this creates a good setting for further work on developing more effective web application security tools based on navigating illustrative and complex websites.

**Strengths:**

The approach is very good. It is useful to have a systematic way to evaluate web application vulnerability tools. The testing setup is interesting and compelling.

**Weaknesses:**

This would be a stronger paper, of course, if it also included tool improvements, validated within this test environment. That is more work, of course, and likely the next step (and next paper) for this team. However, without some indication that at least some of the limitations of current tools may be overcome, it is not clear whether the benchmark sets an achievable near-term goal, or simply a calibration point for the state of the art.

**Questions:**

Are there actionable insights for developing improved web application security tools? Even if not, the characterization of failure modes for existing approaches may be useful.

---

> ### Author Response · Authors · 2025-11-26
>
> Thank you for your positive and constructive feedback. We have conducted more experiments and added more discussions to the revised manuscript to address your concerns.
>
> ---
>
> > This would be a stronger paper, of course, if it also included tool improvements, validated within this test environment. That is more work, of course, and likely the next step (and next paper) for this team. However, without some indication that at least some of the limitations of current tools may be overcome, it is not clear whether the benchmark sets an achievable near-term goal, or simply a calibration point for the state of the art.
>
>
> We have conducted experiments where we disabled the usage of all Kali tools. The results show decreasing performance across all models, confirming that the tools are indeed beneficial for the agents to complete the tasks. We have added these results to the revised manuscript.
>
>
> | Model                    | With Tool (%) | Without Tool (%) |
> |--------------------------|---------------|-------------------|
> | GPT-4o                   | 0.0           | 0.0              |
> | Qwen3-VL-235B-A22B-Thinking  | 0.0           | 0.0              |
> | Claude 4 Sonnet          | 0.0           | 0.0              |
> | Gemini Pro               | 5.6           | 2.8              |
> | GPT-5                    | 8.3           | 5.6              |
> | Claude 3.7 Sonnet        | 11.1          | 5.6              |
>
> ---
>
> > Are there actionable insights for developing improved web application security tools? Even if not, the characterization of failure modes for existing approaches may be useful.
>
> As discussed in Section 4.3 (moved from Section 6 in the original manuscript for greater visibility), we have categorized several common failure modes observed during our experiments. Overall, current CUAs do not show impressive abilities in solving web security tasks, with their failure modes including tool misuse, poor error recovery, and failure to deeply investigate the target system etc.  Appendix B.2.1 further lists the full annotated errors. We believe these insights provide a comprehensive understanding of current limitations and can guide future research directions.
>
>
> **Insight into designing security tools**
> We would like to respectfully note that a discussion on designing security tools is out of the scope of this paper, as our focus is to benchmark the current capabilities of CUAs in web security tasks. However, we agree that specialized tools are designed for expert human users and are not well-optimized for CUA agents. Some tools list all parameters on the screen and expect users to configure them manually, which can be challenging for agents. We believe that adapting existing security tools for agent use is an important direction for future research, and we encourage further exploration in this area.

---

### Official Review · Reviewer_ap1v · 2025-10-31

**Soundness:** 3
**Presentation:** 3
**Contribution:** 3
**Rating:** 6
**Confidence:** 3

**Summary:**

Hackworld curates 36 CTF challenges, sourcefrom NYU ctf bench, CyBench amd InterCode-CTF.
36 web applications over 7 programming language, 11 web frameworks.
Each web app deployed as isolated docker.

The novelty is wrapping them into a CUA-problem.
Environment is Kali OS with security analysis tools, such as Burp Suite, DirBuster, Nikto.

Progress monitored: filesystem operations, tool invocations, HTTP requests.


They use Claude sonnet and Opus CUA agents + UI-Tars and Qwen-VL
0% success rate for open-source models.

Questions:
3) SOM: is that just the SOM or is it combined with Screen shot / Screenshot+AxTree?
I think it does not make sense to test it in isolation and should rather be used to complement Screenshot+AxTree.

**Strengths:**

Realistic attacker environment, usage of industry-standard tools and Kali OS.
Great discussion and insightful work on failure analysis.

**Weaknesses:**

Results are 0% with open-source models. A bit surprising to me given that if we look at result tables from papers like Cybench (Llama, Mixtral) the open weight models  achieve better than 0% success rate.
However, I do see that NYU CTF shows 0% for the open weight models.
Given that you state that choice of observation format is not that important, i would like a discussion why models that succeed in Cybench fail in HackWorld.

**Questions:**

Exploitation rate below 12% + security oblivious behavior -> Can you explain more why this s concerning?
If a computer-use agent fails to capture the flag, it might mean it is less susceptible to cause damage under attacks, e.g., prompt injections, which is not necessarily a bad thing.

Does the CUA agent use the terminal? If so, is it helpful? Does it make sense to restrict direct terminal usage? What are the benefits of CUA vs. terminal-use agent in this context? Do any of the tasks actually require, e.g., very interactive  browser interaction. I would love to see a discussion on this, because using CUA for security-testing vs. a Terminal-use agent is not an obvious choice to me.

---

> ### Author Response · Authors · 2025-11-26
>
> We thank the reviewer for their positive assessment and for recognizing the strengths of HackWorld. We address your specific questions regarding open-source performance, agent design, and safety implications below.
>
>
> ---
>
> > Exploitation rate below 12% + security oblivious behavior -> Can you explain more why this is concerning? If a computer-use agent fails to capture the flag, it might mean it is less susceptible to cause damage under attacks, e.g., prompt injections, which is not necessarily a bad thing.
>
>
> We appreciate the reviewer for identifying this potential ambiguity. We have updated the Abstract and the Introduction for clearer presentation. Essentially, we aim to highlight that current agents struggle with web security tasks in HackWorld, establishing a rigorous baseline for tracking the evolution of agentic capabilities.
>
> ---
>
> > Does the CUA agent use the terminal? If so, is it helpful? Does it make sense to restrict direct terminal usage? What are the benefits of CUA vs. terminal-use agent in this context? Do any of the tasks actually require, e.g., very interactive browser interaction. I would love to see a discussion on this, because using CUA for security-testing vs. a Terminal-use agent is not an obvious choice to me.
>
> We packed the full Kali OS image for the CUA to use (described in Section 2.2) without any restrictions, so the agent could use the terminal to perform basic operations (e.g., curl, ping). It reflects human pentesters' behavior. However, we argue that restricting agents to a terminal-only interface would fundamentally limit the validity of the evaluation for modern web security. Modern applications heavily rely on client-side rendering and dynamic DOM manipulation. A terminal agent viewing raw HTML and JavaScript code must effectively simulate the browser's rendering engine in its context window to understand the application state, which is often impractical and error-prone.
> In contrast, a CUA perceives and interacts with the rendered UI, which allows them to navigate and handle dynamic contents and utilize visual cues that are often essential for effective penetration testing. Such abilities are often critical in creating carefully manipulated states that expose and exploit vulnerabilities, and are required by many HackWorld tasks.
>
> ---
>
> > SOM: is that just the SOM or is it combined with Screen shot / Screenshot+AxTree? I think it does not make sense to test it in isolation and should rather be used to complement Screenshot+AxTree.
>
> We appreciate the reviewer’s interest in the details of the experiments. We agree that SoM must be used in conjunction with the screenshot, and this is exactly how we implemented it in our experiments. We have updated the notation to "Screenshot + SoM" in the revised manuscript to clarify this point.
>
> ---
>
> > Results are 0% with open-source models. A bit surprising to me given that if we look at result tables from papers like Cybench (Llama, Mixtral) the open weight models achieve better than 0% success rate. However, I do see that NYU CTF shows 0% for the open weight models. Given that you state that choice of observation format is not that important, i would like a discussion why models that succeed in Cybench fail in HackWorld.
>
> We thank the reviewer for highlighting this interesting contrast. The performance discrepancy stems from the fundamental difference in interaction modality. CyBench evaluates general security capabilities across a broad spectrum of tasks (such as reverse engineering and crypto), many of which can be solved via text-based terminal scripts. In contrast, HackWorld focuses specifically on web application security, a domain that inherently requires heavy visual perception and GUI interaction, as we explained in Q2. As a result, models that excel in CyBench may fail in HackWorld due to their inability to effectively interpret and interact with complex web UIs.

---

### Official Review · Reviewer_2Tpq · 2025-11-01

**Soundness:** 2
**Presentation:** 2
**Contribution:** 2
**Rating:** 4
**Confidence:** 4

**Summary:**

This paper introduces HackWorld, a new benchmark for evaluating computer-use agents (CUAs) in realistic web environments containing exploitable vulnerabilities. Instead of testing on clean, simplified webpages, HackWorld embeds agents in 36 real vulnerable web applications (built with 7 languages and 11 frameworks) and measures their ability to identify and exploit vulnerabilities using a CTF (Capture-The-Flag) formulation.

**Strengths:**

The paper provides a realistic and technically comprehensive benchmark, incorporating genuine vulnerabilities and real-world tools rather than synthetic setups.

The evaluation framework is well-implemented and could be useful for testing future computer-use agents.

The experimental setup is clear, and the findings provide meaningful insights into the current limitations of CUAs in handling realistic exploitation tasks.

**Weaknesses:**

The motivation is weak and not well justified — it is unclear why evaluating “common-sense defense” or hacking ability is of substantial research value or relevance to practical LLM applications.

 The abstract and introduction are wordy and difficult to follow, which obscures the main contribution.

The conceptual novelty is limited; the work is mostly an engineering effort to build a testbed rather than a new research idea. Moreover, the experiments lack depth in analysis, focusing only on success rates without exploring why models fail or what reasoning gaps cause the failures.

Finally, the evaluation of closed-source models is incomplete — only the Claude family is tested, which weakens the generality of the conclusions.

**Questions:**

see weakness

---

> ### Author Response · Authors · 2025-11-26
>
> We thank the reviewer for their constructive feedback and for recognizing HackWorld as a "realistic and technically comprehensive benchmark" with a "well-implemented" evaluation framework. We are encouraged that you find our findings regarding the limitations of current CUAs to be meaningful.
>
>
> We have addressed your concerns regarding motivation, novelty, and experimental breadth below.
>
> ---
>
> > The motivation is weak and not well justified — it is unclear why evaluating “common-sense defense” or hacking ability is of substantial research value or relevance to practical LLM applications.
>
>
> We respectfully emphasize that evaluating web security skills is essential for automated red teaming and penetration testing, which are central to modern cybersecurity practice. As noted in the Introduction (Lines 43–45), traditional penetration testing requires specialized human expertise and significant manual effort, making it costly and fundamentally unscalable for today’s rapidly expanding web ecosystem. In contrast, agent-based automated defense offers a promising path toward scalable, continuous, and human-free security assessment. Developing agents capable of executing realistic penetration-testing workflows can substantially improve our ability to detect and mitigate vulnerabilities before they are exploited in production.
>
>
> HackWorld directly supports this goal by providing a rigorous benchmark for assessing and advancing the security-reasoning capabilities of LLM-based agents. By grounding evaluation in complex, real-world web vulnerabilities, HackWorld contributes to the development of more robust automated defense mechanisms.
>
>
> ---
>
> > The abstract and introduction are wordy and difficult to follow, which obscures the main contribution.
>
> We appreciate the reviewer’s suggestions for improving clarity and presentation. We have revised the abstract and introduction to enhance the clarity of our motivation.
>
> ---
>
> > The conceptual novelty is limited; the work is mostly an engineering effort to build a testbed rather than a new research idea. Moreover, the experiments lack depth in analysis, focusing only on success rates without exploring why models fail or what reasoning gaps cause the failures.
>
>
> **Novelty:**
> We believe HackWorld contributes significant conceptual novelty by defining a new paradigm distinct from existing works. While curating a reproducible testbed is indeed a rigorous and heavy engineering challenge, the core novelty of Hack World is conceptual. Previous benchmarks like CyBench evaluate penetration testing with text-only LLM models or agents. HackWorld is the first to systematically evaluate vision-based CUAs in this domain (Lines 46-49). This shift to visual interaction is non-trivial, as modern web applications heavily contain complex graphical interfaces and dynamic content that text-only models cannot effectively handle. By enabling agents to perceive and interact with web UIs, HackWorld opens new avenues for research into how MLLMs can understand and manipulate complex visual environments for security tasks.
>
>
> **Error Analysis:**
> Regarding the failure mode analysis, we would like to direct attention to Section 4.3 (which was moved from the Discussion from the original manuscript for greater visibility) for a high-level failure mode categorization and Appendix B.2.1 (unchanged in the revision) for the full annotated results. We identified 8 general failure modes, including tool misuse, poor error recovery, and failure to deeply investigate the target system, etc. We believe these insights provide a comprehensive understanding of current limitations and can guide future research directions.
>
>
> ---
>
>
> > Finally, the evaluation of closed-source models is incomplete — only the Claude family is tested, which weakens the generality of the conclusions.
>
>
> We extended our evaluation to include the strongest available frontier models, specifically GPT-5, Gemini 2.5 Pro, GPT-4o, and the recently released Qwen3-VL-235B-A22B-Thinking. We list the full results below, and compare it with the scores on OSWorld (a benchmark for common computer use) to explicitly demonstrate the unique challenges HackWorld presents.
>
>
> | Model | HackWorld Success Rate | OSWorld Success Rate |
> | :--- | :---: | :---: |
> | | Closed Models | |
> | Claude-4-Sonnet | 0.0 | 43.9 |
> | GPT-4o | 0.0 | 5.0 |
> | Claude-4-Opus | 0.0 | - |
> | Claude-3.5-Sonnet | 2.8 | 14.9 |
> | Gemini Pro | 5.6 | - |
> | GPT-5 | 8.3 | - |
> | Claude-3.7-Sonnet | 11.1 | 27.1 |
> | | Open Models | |
> | Qwen2.5-VL-72B-Instruct | 0.0 | 5.0 |
> | UITARS 1.5 (7B) | 0.0 | 27.3 |
> | Qwen3-VL-235B-A22B-Thinking | 0.0 | 38.1 |
>
>
> Our conclusions from the original submission remain consistent: general computer use capabilities struggles to  perform adversarial exploration and specialized tool usage; more recent models do not necessarily achieve better results on HackWorld; and open-source CUAs still lag significantly behind closed ones on these tasks.

---

### Official Review · Reviewer_fyMi · 2025-11-01

**Soundness:** 3
**Presentation:** 3
**Contribution:** 3
**Rating:** 4
**Confidence:** 3

**Summary:**

The paper proposes HackWorld, a GUI-based benchmark for evaluating Computer-Using Agents (CUAs) that try to discover and exploit real web-application vulnerabilities. The benchmark aggregates 36 containerized CTF-style challenges spanning multiple frameworks and languages, runs in a Kali-based environment with industry-standard security tools, and evaluates success via flag capture with reproducible logs. The study shows that current CUAs perform poorly (<~12% success), analyzes failure modes (planning, tool use, and GUI manipulation), and argues that “web hacking via GUI” is substantially harder than generic web browsing or visual automation.

**Strengths:**

- The task is clearly defined and the CTF style evaluation metric is unambiguous.
- Using MLLM-based agents to autonomously discover web-application vulnerabilities is a hard and meaningful task.
- The paper provides detailed analysis on the error cases, which can guide future research.

**Weaknesses:**

- Backbone diversity is narrow. Only six models are tested, four from the same family (Claude), plus a single 7B model that is too weak for this task. This limits the generality of the conclusions. Stronger open and closed models, and more families, are needed.
- While the curation is solid, 36 tasks is still modest for a general benchmark.
- As a benchmark paper, it is unclear what specific new agent capabilities are evaluated compared to OSWorld [1].

[1] Xie, Tianbao, et al. "Osworld: Benchmarking multimodal agents for open-ended tasks in real computer environments.", 2024

**Questions:**

- Could you report results on stronger models such as GPT and Gemini, which have demonstrated strong performance in the OSWORLD paper, and compare them with human results?
- Compared with OSWorld, does HackWorld evaluate a fundamentally different type of agent capability, or is the distinction mainly in the cybersecurity domain knowledge? What new abilities or reasoning skills are specifically required of agents in this setting?

---

> ### Author Response · Authors · 2025-11-26
> **More backbones and comparisons with OSWorld**
>
> We thank the reviewer for recognizing the clear definition, meaningful task design, and detailed error analysis of HackWorld. We appreciate the constructive feedback on backbone diversity and task scale. Below, we address your questions with new experiments and clarified design choices.
>
> ---
>
> > Could you report results on stronger models such as GPT and Gemini, which have demonstrated strong performance in the OSWORLD paper, and compare them with human results?
>
>
> We extended our evaluation to include the strongest available frontier models, specifically GPT-5, Gemini 2.5 Pro, GPT-4o, and the recently released Qwen3-VL-235B-A22B-Thinking. We list the full results below to explicitly benchmark HackWorld against top performers in OSWorld where available. The result has been added to the Appendix.
>
>
> | Model | HackWorld Success Rate | OSWorld Success Rate |
> | :--- | :---: | :---: |
> | | Closed Models | |
> | Claude-4-Sonnet | 0.0 | 43.9 |
> | GPT-4o | 0.0 | 5.0 |
> | Claude-4-Opus | 0.0 | - |
> | Claude-3.5-Sonnet | 2.8 | 14.9 |
> | Gemini Pro | 5.6 | - |
> | GPT-5 | 8.3 | - |
> | Claude-3.7-Sonnet | 11.1 | 27.1 |
> | | Open Models | |
> | Qwen2.5-VL-72B-Instruct | 0.0 | 5.0 |
> | UITARS 1.5 (7B) | 0.0 | 27.3 |
> | Qwen3-VL-235B-A22B-Thinking | 0.0 | 38.1 |
>
>
> These results suggest that **general computer use capabilities do not linearly generalize to cybersecurity tasks**. While models like Claude-4-Sonnet excel at linear tasks in compliant environments (e.g., spreadsheet processing), they struggle with the adversarial exploration, reasoning and specialized tool usage for exploitation, which are required in HackWorld. Conversely, Claude-3.7-Sonnet achieved the highest score (11.1%) due to higher stability in the usage of web security tools. All open source models we benchmarked failed to demonstrate any meaningful attempts on HackWorld, despite some showing moderate performance on OSWorld.
>
>
> We emphasize that reporting a single, aggregated human baseline for HackWorld is not meaningful due to the extreme variance in human capabilities. Performance on HackWorld depends heavily on reasoning with specialized expertise and exploiting skills rather than general coding ability. Consistent with precedent in related benchmarks such as InterCode, CyBench, and NYU CTF Bench, we therefore do not report a human score.
>
> ---
>
>
> > While the curation is solid, 36 tasks is still modest for a general benchmark.
>
>
> During data collection, we manually sourced 473 web-related challenges from global CTF competitions (2019–2025) using public CTF archives. However, none of these 473 challenges satisfied the strict quality requirements necessary for a rigorous, fully automated benchmark. Our verification uncovered substantial usability issues that made the tasks unsuitable for reproducible agent evaluation: 103 lacked Docker configurations, 118 experienced Docker build timeouts, and 62 eventually failed to build. Among the subset that did build, 162 suffered from runtime errors or broken dependencies, and others lacked a verifiable flag. Because these issues make automatic evaluation fundamentally unreliable, we finalized our dataset using only 36 high-quality, fully reproducible challenges.
>
>
> It is also important to contextualize the scale of previous works. Cybench contains 40 tasks, but its larger count stems from including more non-web challenges that do not require visual interaction or GUI operations. Given the difficulty of identifying stable, containerized, visually interactive web tasks, our final set of 36 challenges is fully aligned with the standard in this high-complexity domain.
>
>
> ---
>
> > Compared with OSWorld, does HackWorld evaluate a fundamentally different type of agent capability, or is the distinction mainly in the cybersecurity domain knowledge? What new abilities or reasoning skills are specifically required of agents in this setting?
>
>
> HackWorld evaluates a fundamentally different paradigm than OSWorld. While OSWorld benchmarks daily common tasks within compliant environments, **HackWorld demands capabilities rooted in adversarial exploration, reasoning, and specialized web security tool usage for exploitation**. Unlike the explicit goals and deterministic paths of OSWorld, HackWorld forces agents to operate under severe information asymmetry, actively probing resistant systems to reveal hidden attack surfaces. For instance, encountering an HTTP 403 error in HackWorld requires an agent to deduce the restriction, hypothesize alternative vectors (e.g., exploiting misconfigurations), and orchestrate tools to verify them. This stands in sharp contrast to OSWorld, where agents primarily rely on linear instruction following and planning to achieve transparent goals.

---

### Author Response · Authors · 2025-11-26

We thank all reviewers for their detailed reviews. We have made the following updates to the paper with the additional blue text.

**Addition**: Improved presentation of evaluation performance of CUAs in Abstract on Line 30, as asked by Reviewer 2Tpq.

**Replace**: Clarification of the ability required by Hackworld in the Introduction on Line 95, as asked by Reviewer fyMi.

**Addition**: Highlight error analysis by moving it from Discussion to Section 4.3, as asked by Reviewer 2Tpq and mq7d.

**Replacement**: Specifying “Set-of-Marks” into “Screenshot + Set-of-Marks ”, as asked by Reviewer ap1v.

**Addition**: The comparison of success rates of computer-use agents on HackWorld and OSWorld benchmarks in Section 4.4, as asked by Reviewer fyMi.

**Addition**: The performance comparison of computer-use agents with and without tools in Appendix B.2.3, as asked by Reviewer mq7d.

Furthermore, we sincerely ask the reviewers to check out the Appendix for more details, though it might be a bit lengthy. It may already address some of your concerns.

We appreciate the positive sentiment expressed about HackWorld, and are very excited about future work utilizing and building on HackWorld. We will continue to maintain the benchmark and the leaderboard.

---

### Author Response · Authors · 2025-12-04
**Summary for Area Chair**

Due to the rollback, we briefly summarize the rebuttal process here. All ongoing discussions occurred during the data leakage incident.

- We have added **all the experiments** asked by **Reviewer fyMi**, **Reviewer 2Tpq**, and **Reviewer mq7d**.

- We have **adequately answered** the questions asked by all the reviewers. Due to the incident, we did not receive any responsed from the reviewers.

- Since **Reviewer fyMi** and **Reviewer 2Tpq** did not fully understand this direction, they had some misunderstandings about our work. We have further addressed their previous concerns.

---

### Meta-Review · Area_Chair_yBaY · 2026-01-05

**Summary:**

This paper introduces HackWorld, a benchmark for evaluating computer-use agents (CUAs) on realistic web application exploitation tasks via graphical user interfaces. The benchmark consists of 36 carefully curated, containerized web applications spanning multiple languages and frameworks, each containing real-world vulnerabilities. HackWorld evaluates agents using a Capture-the-Flag (CTF) paradigm in a full Kali Linux environment with standard security tools, enabling systematic assessment of visual interaction, adversarial exploration, and tool usage.

Reviewers' concerns include:
(1) Backbone model diversity
(2) Limited benchmark scale (36 tasks)
(3) Comparison and differentiation from OSWorld
(4) Conceptual novelty vs. engineering contribution
(5) Depth of failure analysis beyond success rates
(6) Lack of tool improvement
(7) Unexpected zero performance of open-source models

After the rebuttal, concerns were addressed.
For (1), the authors added evaluations on additional backbone models.
For (2), the task scale is considered reasonable given the difficulty and curation effort.
 For (3), HackWorld clearly differs from OSWorld by focusing on adversarial web vulnerability exploitation rather than general computer-use tasks.
For (4), AC feels that this paper is primarily a benchmark and  represents a meaningful addition to the web security and CUA evaluation domain, it is a good contribution to the domain.
For (5) and (7), the authors provided additional analysis and explanations.
For (6), new experiments were conducted to study the impact of tool usage.

**Reviewer Concerns:**

Reviewers' concerns include:
(1) Backbone model diversity
(2) Limited benchmark scale (36 tasks)
(3) Comparison and differentiation from OSWorld
(4) Conceptual novelty vs. engineering contribution
(5) Depth of failure analysis beyond success rates
(6) Lack of tool improvement
(7) Unexpected zero performance of open-source models

After the rebuttal, concerns were addressed.
For (1), two reviewers pointed out this concern. the authors added evaluations on additional backbone models.
For (2), AC thinks the task scale is considered reasonable given the difficulty and curation effort.
 For (3), HackWorld clearly differs from OSWorld by focusing on adversarial web vulnerability exploitation rather than general computer-use tasks. This concern is not valid.
For (4), AC feels that this paper is primarily a benchmark and  represents a meaningful addition to the web security and CUA evaluation domain, it is a good contribution to the domain.
For (5) and (7), the authors provided additional analysis and explanations.
For (6), new experiments were conducted to study the impact of tool usage.

**Reviewer Scores:**

Reviewer fyMi and 2Tpq will increase their score to positive since authors have addressed their concern well in rebuttal.
Other reviewers will maintain positive for this paper.

---

### Decision · Program_Chairs · 2026-01-26

Accept (Poster)